# What Happened to the Forests of Sierra Leone?

**Richard A. Wadsworth** [1,*] and **Aiah R. Lebbie** [2]

[1]    Department of Biological Sciences, Njala University, Njala, Sierra Leone
[2]    National Herbarium of Sierra Leone, Department of Biological Sciences, Njala University, Njala, Sierra Leone; alebbie@njala.edu.sl
[*]    Correspondence: rwadsworth@njala.edu.sl; Tel.: +232-(0)-79-675-004

**Abstract:** The last National Forest Inventory of Sierra Leone took place more than four decades ago in 1975. There appears to be no legal definition of "forest" in Sierra Leone and it is sometimes unclear whether reports are referring to the forest as a "land use" or a "land cover". Estimates of forest loss in the Global Forest Resource Assessment Country Reports are based on the estimated rate during the period 1975 to 1986, and this has not been adjusted for the effects of the civil war, economic booms and busts, and the human population doubling (from about three million in 1975 to over seven million in 2018). Country estimates as part of the FAO (Food and Agriculture Organization of the United Nations) Global Forest Assessment for 2015 aggregate several classes that are not usually considered as "forest" in normal discourse in Sierra Leone (for example, mangrove swamps, rubber plantations and Raphia palm swamps). This paper makes use of maps from 1950, 1975, and 2000/2 to discuss the fate of forests in Sierra Leone. The widely accepted narrative on forest loss in Sierra Leone and generally in West Africa is that it is rapid, drastic and recent. We suggest that the validity of this narrative depends on how you define "forest". This paper provides a detailed description of what has happened, and at the same time, offers a different view on the relationship between forests and people than the ideas put forward by James Fairhead and Melissa LeachIf we are going to progress the debate about forests in West Africa, up-to-date information and the involvement of all stakeholders are needed to contribute to the debate on what to measure. Otherwise, the decades-old assumption that the area of forest in Sierra Leone lies between less than 5% and more than 75%, provides an error margin that is not useful. This, therefore, necessitates a new forest inventory.

**Keywords:** Sierra Leone; forest; land use; land cover; mapping; remote sensing; forest loss

## 1. Introduction

According to the 2015 Sierra Leone Country Report prepared as part of the FAO (Food and Agriculture Organization of the United Nations) Global Forest Assessment [1] there has been no national forest inventory since 1975. An estimate of change was made for the period 1975 to 1986 [2], but without an inventory it is unclear how this was done. The rate of forest loss reported in the statistics, since then (30,000 ha per year) is simply the 1975–1986 rate, with no account taken of population growth (from three million to seven million), economic conditions or the civil war. The civil war may have displaced 2.6 million people which was more than half the population at that time [3]. Changes in the distribution of population, due to the war are known to have caused changes in land cover and an assessment of forest loss by Chiefdom had a negative correlation with measures of armed conflict [4].

The accepted wisdom of forest loss in Sierra Leone is that it is recent, rapid and drastic, the validity of this depends in part on how "forest" is defined. While the dominant narrative is not supported by the data we have available, it is also not particularly supportive of the ideas put forward by Fairhead and [5–7] which again depend on exactly what is meant by forest. These questions of semantics are

critical if progress is going to be made in the debate about forests in West Africa. At the moment, it is possible to defend saying the area of forest in Sierra Leone is anywhere between less than 5% and more than 75%, and such an error margin is really not very useful.

Most objects in physical geography may be considered "uncertain" objects in a technical sense. While the uncertainty may be a simple matter of measurement error, more often it is because the definition of the object is either "vague" or "ambiguous" and if it is ambiguous it may be either "not specific" or "contested" [8]; often the problem is related to Sorites paradox; for example, we may all agree that one tree does not constitute a forest, nor does two, but maybe a group of 1,000 trees represents a forest? But if I cut one tree so there are only 999 is it still a forest? In such a situation any threshold or boundary can, quite rightly, be contested and the concept of the forest becomes "uncertain". The problem is particularly acute in the case of land cover where terms are used as if everyone was using the same "model" but different actors may be applying very different definitions for wholly legitimate reasons [9,10]; in the case of "forest" the term is further complicated in that it may apply to either a "land cover" or a "land use" [11].

In the draft Forest Policy for Sierra Leone 2010 [12], the forest is not defined, but the area of forest is given as 5%, which is exactly the same percentage as given for the "closed high forest" class in 1975. The FAO Global Forest Resource Assessment Report for Sierra Leone for 2010 gives a slightly smaller area of forest (equivalent to about 3.6%) but in the country report of 2015 [1] the forest cover is given as 14.7%, with other wooded land at 61.1%. This is at least partially a definitional issue as the 2015 figure is an aggregation of several land cover classes; closed high forest, secondary forest, coastal woodland, fringing swamp forest, mangrove, Raphia palm swamp, rubber plantations (but not oil palm), savanna woodland (and *Lophira* savanna and mixed species savanna are "other woodland"). Forests in Reference [1] also include 50% of the "forest regrowth" (farm bush) where it is not dominated by agriculture (but it is unclear how fallow can be considered as not part of an agricultural system). The history of forest policy in Sierra Leone is discussed in detail in two PhD theses [13,14] which attempt to place the development of stated and implemented policies (which often conflict) in the context of colonialism and more recently pressure from INGOs (International Non-Governmental Organizations) and donors.

There is a considerable literature about Freetown (founded in 1787), but the original "Sierra Leone" was restricted to what is now the "Western Area"; the Northern, Southern and Eastern provinces (which are more than 99% of the total area of modern Sierra Leone) was only declared a "protectorate" in 1896 (to protect trade routes into Freetown and counter French expansion in Guinea). As such the literature on the Protectorate is much more limited than that of Freetown. There are two early reports on the forests of the Protectorate; 1909 and 1911 [15,16] (reported in Reference [17]) which state that by that time "forest" (closed canopy rain forest) covered as little as 1% of the land area. The "dominant narrative" of forest loss in Sierra Leone is that the country was once dominated by closed canopy rain forest in the 1900s although some authorities who are possibly not aware of [15] or [16] and suggest that extensive loss of forest started in the 1920s [18], 1940s [19] or the 1960s [20], but all suggest a very rapid deforestation primarily caused by poor farmers. A less common contributing factor is the export of timber from the provinces and while it is known to have been important in the 19th Century, its extent and impact on land cover are debated; [21,22] consider that it may, in some locations have been extensive. The direct and indirect impact of the railway has also been suggested as a major but localized cause of deforestation [19,22]. The perception that Sierra Leone was once heavily forested may be the result of observing that more than 50% of the country is covered in "bush-fallow" vegetation and that if this fallow is left undisturbed, the trees will continue to grow and the species composition gradually change. Indeed, it is still quite common to see bush-fallow described as "forest regrowth." However, the tree species composition of the bush-fallow is relatively impoverished and it is dominated by early pioneer tree species that coppice freely and vigorously. Theoretically places like Tiwai Island will allow us to record the gradual transition from bush-fallow to the forest, as farming ceased on the Island in the 1980s. There is some recent (2015) very detailed airborne LiDAR data for a

small part (16 km²) of Gola [23], which indicates that the forest subjected to selective logging is still not structurally identical to unlogged forests after many decades. The implication is that a transition from farm-bush to forest is a very long process. In an extensive series of papers [5–7] review the history of the "forest islands" of the Guinea savanna and demonstrate that contrary to early descriptions, they are not remnants of a once extensive forest but the result of human interventions in reducing fires and introducing or encouraging economically valuable trees. Reference [5] has extended the argument to the more humid environment of Sierra Leone (among other countries) to argue that the country has, at least for a very long time, been a mosaic of vegetation types, with forests occupying only a relatively small proportion of the area. Similarly, Reference [24] identify the positive role of local communities in maintaining and developing sustainable productive landscapes. The question of when, if ever, the country was covered in closed-canopy rain forest is subject to considerable uncertainty, after reviewing the available evidence [17] conclude "*it is simply not known* how much of Sierra Leone was once covered by primary forest" (their emphasis).

　　Forest is defined by the FAO as an area of land greater than 0.5ha, with a tree canopy covering more than 10% of the area and trees greater than five meters high [1]. (Note that other parts of the UN use different definitions). Definitions like this led to all bush-fallow being classified as forest and statements that more than half the country is forest (e.g., Reference [4]) or "other woodland" (e.g., Reference [1]). The FAO definition appears to be a compromise among those countries that define forest in terms of its biophysical properties (see Figure 1). Note that most countries (including Sierra Leone) do not appear in Figure 1, and this is because they define forest in other ways (if they define it at all).

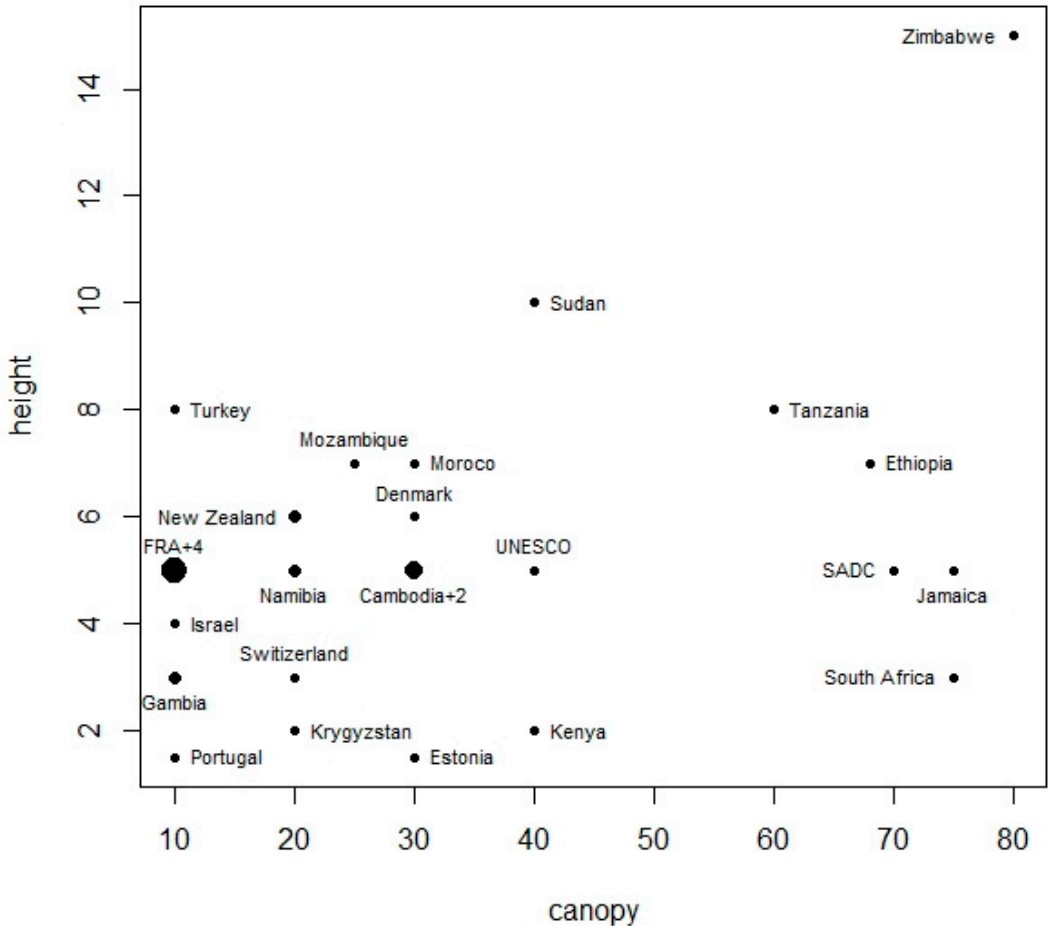

**Figure 1.** Countries that use canopy cover and tree height to define "forest" (values from Reference [25]).

The Global Forest Watch uses MODIS data and a threshold of > 30% tree cover, giving an area of tree cover of 61% in 2001 and a loss of about 32,000 ha per year. Only 7% of this is said to represent a permanent land cover change; loss includes "Improved detection of smallholder rotation agricultural clearing in dry and humid tropical forests." Global Forest Change uses a methodology defined by Reference [26] using Landsat data, where tree cover is estimated as percentage cover of vegetation greater than five meters (although it is difficult to tell how vegetation height can be estimated from multi-spectral data).

Figure 2 provides a schematic of what an FAO forest would look like; this is rather sparser than the common visualization used in Sierra Leone where the "forest of the imagination" is a thick canopy and massive trees, a frightening and awe-inspiring sight.

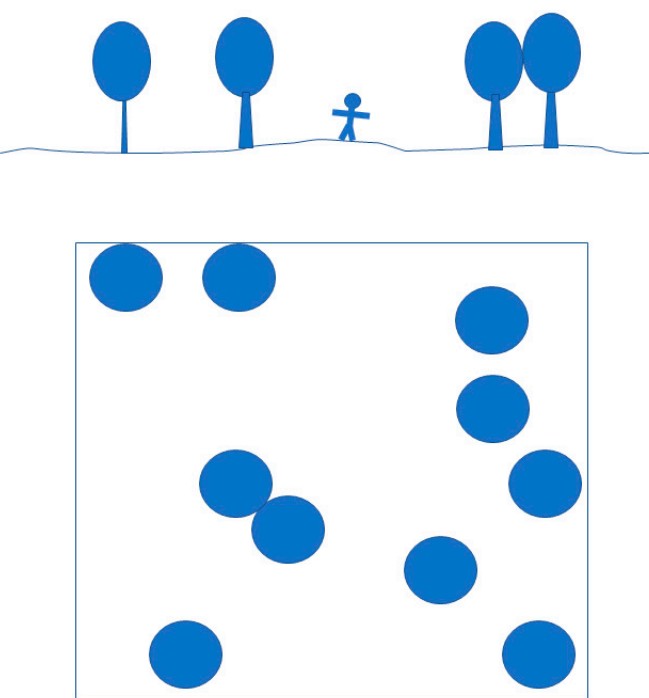

**Figure 2.** Schematic of the Global Forest Resource Assessment definition of a "forest"; area > 0.5 ha, trees > 5 m, canopy cover > 10%.

## 2. Materials and Methods

Three land cover maps have been assessed to consider how the extent of the forest has changed in Sierra Leone over a 50-year period. In addition, a digital map of tree cover based on a per-pixel classification of remote sensing images has been investigated. Each of the three land cover maps exists as a "jpeg" image; unfortunately, the metadata on who scanned them and when has not been included with the images (or if it was once included has now become detached). The images were imported into GoogleEarth and then stretched until the boundary on the images matched as closely as possible to the boundary on GoogleEarth. Note that as we lack information on the original projection, datum and spheroids used we cannot "project" the images and some distortions are possible. Once the images had been fitted to the geographic projection the boundary of the various land units were digitized on screen. The output from the digitizing was then cleaned up in an open source GIS OpenJump [27] to remove duplicates, incomplete polygons and other artifacts and a shapefile generated for mapping. Some processing of the data on tree cover per pixel [26] was done to aggregate from the pixel to parcels and to threshold those parcels where tree cover might constitute "forest" using the R statistical package [28], R was also used for some of the cartography.

The three land cover maps are;

1. "1950" land cover map, this carries the rubric "drawn by the Surveys and Land Department Sierra Leone 1951 from data supplied by the Soil Conservation Team 1950". We have been unable to find any documentary record of the mission or objectives of the Soil Conservation Team. The map has the reference SLS No 211/1951 and the cartography was done in Nigeria. It shows six classes; forest, farm bush, grass, savanna, *Lophira* savanna and mangrove. There is no description of what is meant by forest, but the implication is that it is a land cover term. This map is one of a series with the same rubric that includes population density, length of fallow period, soils, topography, causes of land degradation and planned production zones.

2. "1975" land cover map, titled "Vegetation and Land Use of Sierra Leone" mapped at a nominal scale of 1:500,000 from infrared colour photographs taken in 1975/6 at a scale of 1:70,000. Produced as part of the Land Resources Survey Project SIL/73/002 FAO/UNDP-MANR, there is an accompanying monograph FAO/LRSP Technical Report No 2 on Vegetation and Land use in Sierra Leone [29]. The "minimum mappable unit" (MMU) is in the order of 600 meters across so many mapped regions are mosaics of several classes. One important consequence of the MMU is that it excludes typical Inland Valley Swamps (IVS) are they are not mapped (the larger boli swamps and coastal swamps are mapped).

3. "2002" land cover map, produced from nine Landsat TM multispectral images dating from 2000 to 2002; the associated paper [30], includes measures of relatedness between categories in the form of a dendrogram and between the categories used in the map and those of three major international schemes; IGBP [31], FAO Land Cover [32] and Federal Geographic Data Committee National Vegetation Classification [33]. The original map was a per-pixel classification at the 25-meter resolution but only an image of that data set was available with a resolution comparable to that of the 1975 map. Because of the problem of separating unmanaged mixed plantations from natural forest [30], the authors chose to use the term "tall trees" rather than "forest".

As the most recent land cover map is now nearly two decades old, we investigated assessment forests by examining contemporary images on GoogleEarth (GE), which vary from contemporary to four years old. As GE imagery is restricted to panchromatic imagery with no near or middle infrared information, automatic or semi-supervised classification methods are not possible. Manually delineating forests is possible, although it was beyond our resources to do this consistently for the whole country. There remains the problem of distinguishing the natural forest from mixed plantations where these consist of an uneven canopy. We discuss the options for producing a contemporary land cover map or forest inventory in later sections.

## 3. Results

Figure 3, illustrates the areas defined as "forest" (1950), "primary forest" (1975) and "tall trees" (2002). What is remarkable is how similar each of these are given the 50-year time span, the narrative of rapid and extensive deforestation, the different definitions, cartography, map bases and purposes for which they were mapped. The major difference is the disappearance of the Tonkolili Forest between Makeni and the Loma mountains; in the 1950s this was of a similar magnitude to the Gola Forests but had declined to less than 10% of its original area by 2002 (Table 1). The edges of the other forests have oscillated over time but this partly represents cartographic and definitional issues and partly represents economic development and pressures.

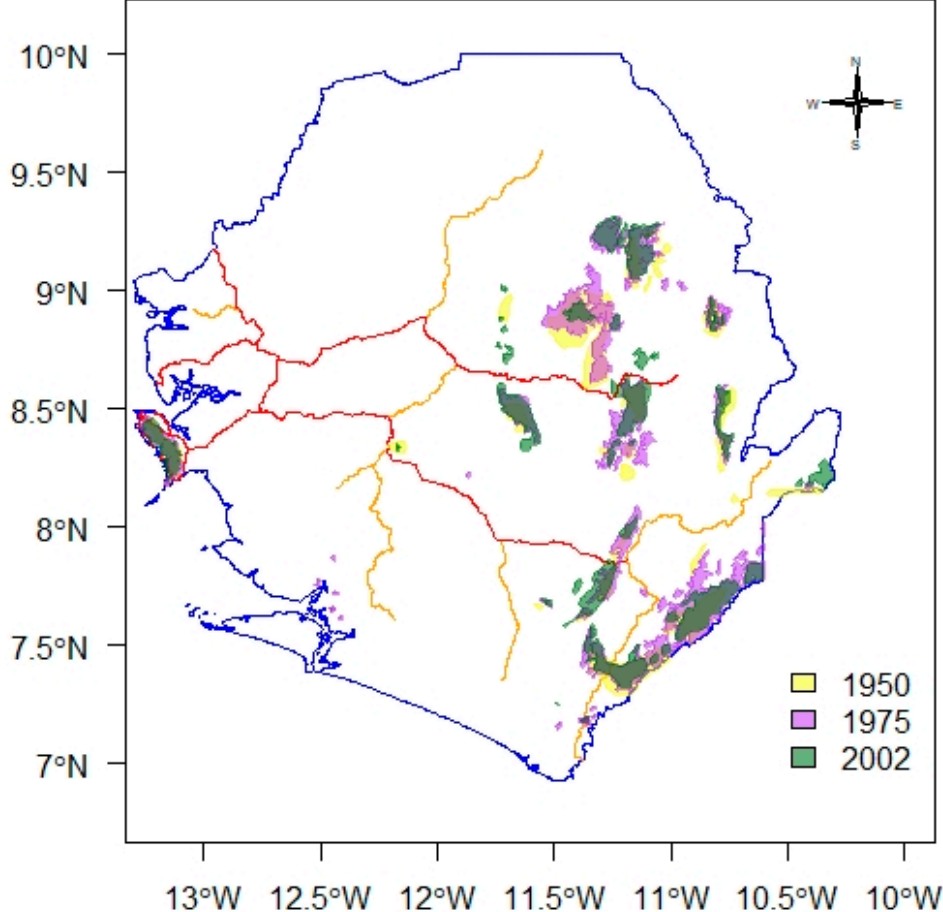

**Figure 3.** Major forested areas in Sierra Leone, 1950, 1975, and 2002. (Major roads indicated in red (tarred) and orange (gravel)).

Table 1 indicates the change in the area of the major forests in Sierra Leone. The figures should be used with caution as some differences are due to different survey methods and cartographic conventions, different definitions and the problems of putting a "crisp" boundary around a "fuzzy" object (in this case a mosaic of habitats).

**Table 1.** Mapped Areas (in sq km) of the larger forest fragments shown on the three maps.

| Mapped Areas | "1950" | "1975" | "2002" |
|---|---|---|---|
| Western Area Forest | 144.6 | 205.8 | 187.1 |
| Gola | 753.5 | 757.1 | 751.4 |
| Kambui | 110.3 | 258.3 | 290.6 |
| Tonkolili | 758.8 | 642.0 | 73.6 |
| Tingi | 119.6 | 88.2 | 34.1 |
| Kangari | 89.4 | 184.5 | 234.4 |
| Nimi | 110.0 | 384.9 | 274.4 |
| Gori | 135.0 | 108.4 | 114.5 |
| Loma | 189.8 | 422.3 | 394.9 |
| "Other" | 457.8 | 675.2 | 243.4 |
| Total mapped | 2679.0 | 3726.7 | 2598.4 |
| % of land | 3.7 | 5.2 | 3.6 |

In the Western Area the early 1950s seems to represent a low point in forest extent. In 1942 a saw mill was established at Number 2 River as part of the war effort [12] but was closed in 1945 when it ran out of logs and the equipment was moved to Kenema. The 1950s map may therefore

represent the physically inaccessible core of the forest. By 1975 the forest edge had expanded, except where it was adjacent to Freetown. By the end of the civil war the forest was starting to rapidly retreat from areas in the north close to Freetown and as the whole strip of land from Freetown to Waterloo started to develop. This trend has continued to the present day with a gradual retreat in the north and east. So far poor access seems to have preserved more of the forest along the western and southern edge, but development and deforestation are evident all along the coast (Figure 4). Munro 2009 [34] reproduces two maps of the forest extent just after the war which show similar patterns to that of [30]. Munro [34] also provides a detailed analysis of the causes of change from a political ecology perspective; noting in particular; (i) the conflicting interests between the Ministry of Lands, Housing, Country Planning and the Environment and the Ministry of Agriculture, Forestry and Food Security, (ii) the disinclination of NGOs and external donors to confront the actions of the urban elite, and (iii) the tacit agreement to concentrate their efforts on the action of the rural poor at the southern end of the peninsula. Mansaray et al. [35] provide more recent data for the northern end of the peninsula indicating a continuing expansion of Freetown from 44.0 to 88.4 km$^2$ between 2001 and 2015. Unfortunately, as they only map the northern half of the peninsular, they do not provide a figure for the whole WAF.

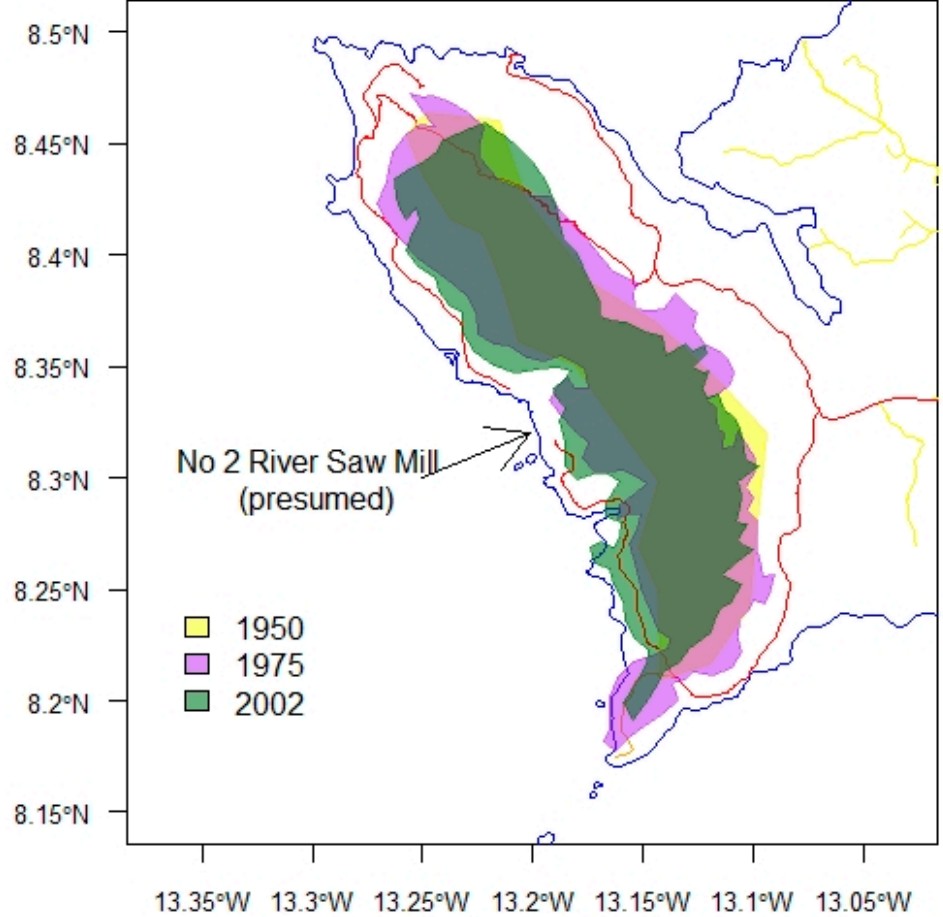

**Figure 4.** The extent of the Western Area Forest, Sierra Leone, 1950, 1975, and 2002. (Major roads in red, but the Peninsular road is not yet completely rehabilitated).

In the south-east of Sierra Leone, the Gola Forest (now a National Park) has retained a forest character despite commercial logging operations in the 1960s and 1970s. There are apparent changes in the edge of the forest, but the extent that these reflect differences in cartography rather than observable changes on the ground is open to debate. During the civil war, pressure on the forest reduced as many people were unwilling to enter the forests, due to fear of rebels and harvesting timber was not an

important activity for the rebels. Parts of Gola have been subject to very detailed analysis by a wide variety of remote sensing techniques. As well as the 16km$^2$ analyzed by Reference [23], for structure and recovery from selective logging, there are papers on the fusion of data from Landsat with radar, Lidar and hyperspectral data for estimating above-ground biomass and landcover [36–38]. The nearby Kambui Hills which are adjacent to Kenema tell a slightly different story. Kenema is the third largest city in Sierra Leone and during the civil war the population rapidly expanded putting a lot of stress on the environment (building materials, firewood, farm land, etc.). These stresses were such that between 1975 and 2002 the northern end of Kambui was deforested while at the southern end the forest expanded, as people left the area or were frightened to enter the forest because of rebels. Since the end of the war the pressure on the northern end has decreased and some recovery has occurred while the resumption of farming in the south has reduced the regrowth (Figure 5).

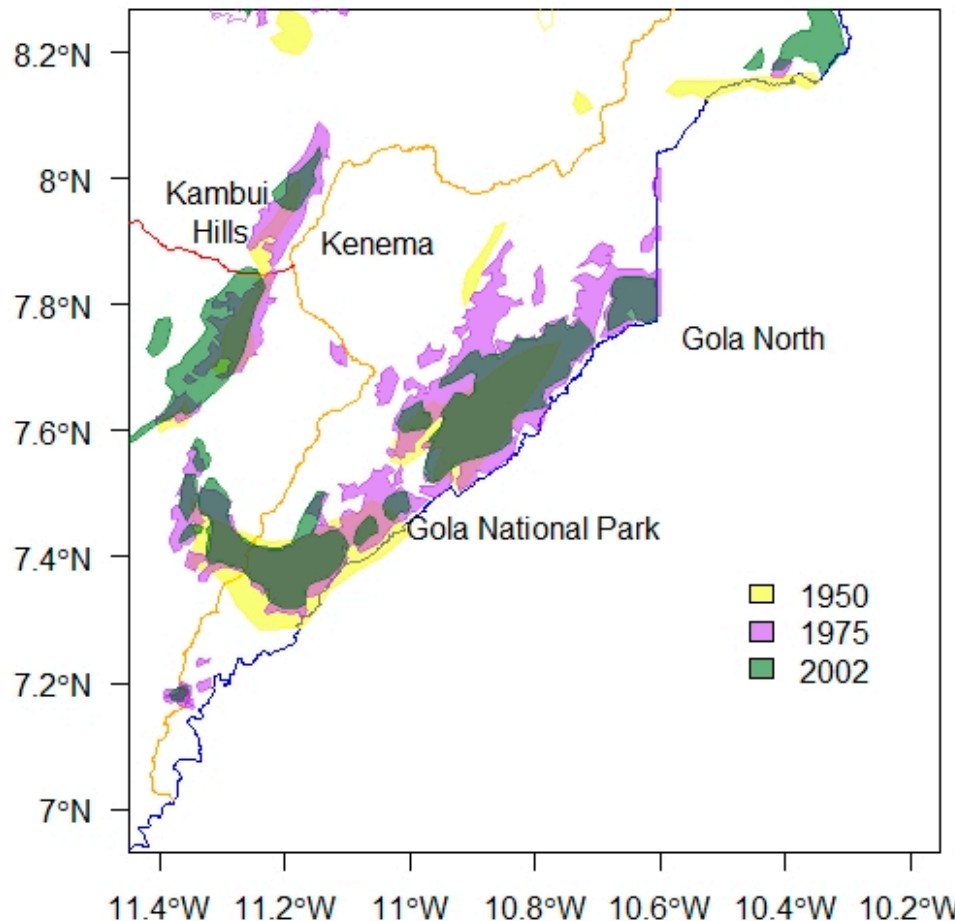

**Figure 5.** Forests of the South-East; Gola and the Kambui Hills.

In the 1950s the forests of Tonkolili was one of the largest areas of forest in Sierra Leone. Since then it has declined to only a small and fragmented area (Figure 6). The reasons why this forest has declined so rapidly is unclear, but most of the former forest is now under bush-fallow farming and it is speculated that this may be as a result of the land being cleared for farming, as well as logged to supply the main diamond mining areas in Kono and the cities of Makeni and Freetown. The forests of Loma, Kangari and Tingi are all under pressure from gold mining but have not declined to anywhere near the same extent. In the case of Loma (Bintumani) commercial logging is minimal as road access is so poor and the terrain makes extraction of boards very difficult. Kangari and Tingi are also probably partly protected by the inhospitable nature of the terrain.

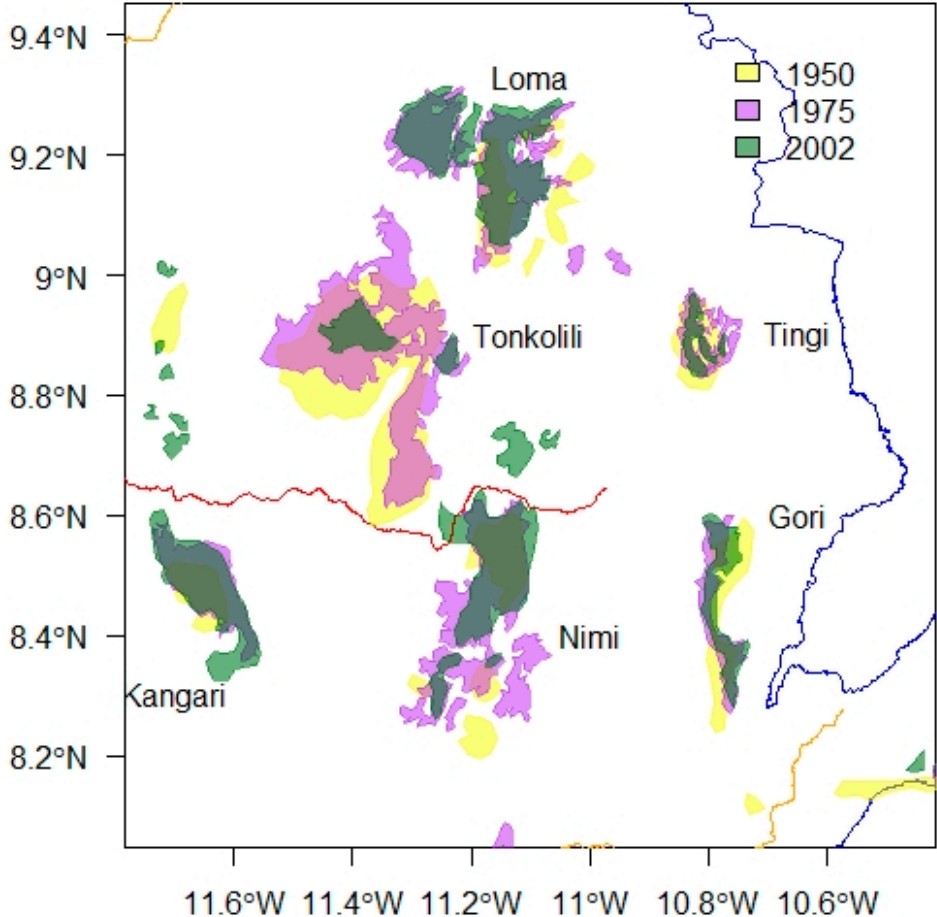

**Figure 6.** Forests of Central Sierra Leone showing the reduced area of forests of Tonkolili.

A comparison has been made between the [26] data tile; Hansen_GFC-2016-v1.4_treecover2000_10N_020W.tif and the estimates from Reference [30]. They both used Landsat data from the year 2000 plus or minus two years, but process it in different ways. The data in Reference [26] consists of the estimated tree cover in each 30-meter pixel (where a tree is "permanent vegetation higher than five meters"), and cells take values between 0 and 100%. For this analysis coverage was converted into a "belief" that a cell (pixel) was "forest" with coverage less than 50% getting a zero, coverage greater than 80% a one and values in-between 50% and 80% a progressively stronger belief (that is taking a "fuzzy logic" approach). The data in Reference [26] is at 30-meter resolution, but in the Sierra Leone context it isn't reasonable to have a forest only 30-meters across; therefore a "focal neighborhood" function was applied to generate an average value of the belief in the neighborhood of each pixel; the search radius was set at 17 pixels, giving an MMU which is comparable with the MMU used in 1975. After performing the neighborhood averaging, those pixels with a mean belief value greater than 0.5 were considered as "forest". Figure 7 compares the two data sets and shows reasonable agreement. The largest differences are around Gola North and the Western Area Forest (WAF). Around Gola the processed data from Reference [26] predicts "forest" while [30] maps "secondary forest" (not shown). In the (WAF) the processed data predicts much less forest, probably due to shadowing in the terrain; Wadsworth et al [30] were forced to create training data for several sub-classes based on how the terrain was illuminated when the Landsat sensor passed over, and after classification they were aggregated into a single class. As [26] is a global data set it is unlikely that they have access to such detailed local intensive collection of training data and this may explain why their estimates of tree cover are relatively low across most of the WAF, (comparable with densities to those of "farm bush" on flat land).

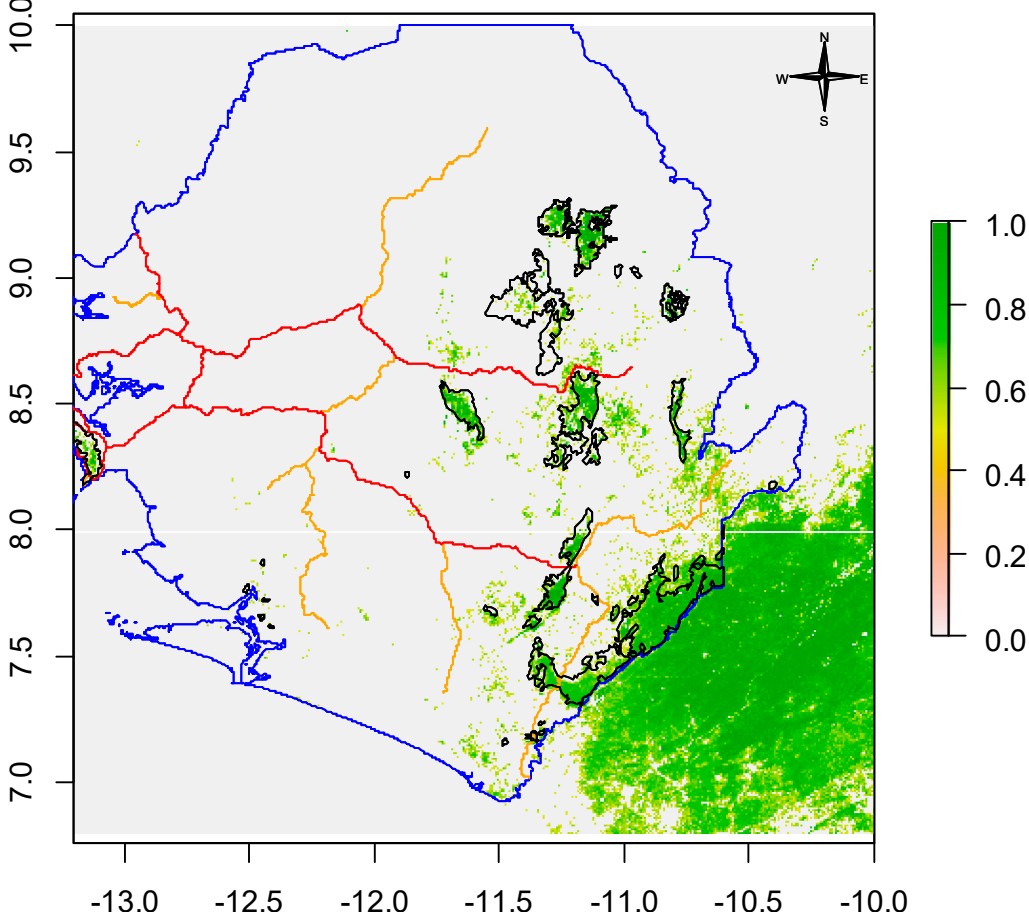

**Figure 7.** A comparison of "forest" cover in 2000 [30] shown with the black outline, and processed from Reference [26] as green infill (see text for details).

One of the most widely used statistics of forests is the FAO Country Reports [1] produced every five years based on information compiled by the relevant national authorities. In Sierra Leone the Country Reports aggregate several classes that are not typically considered forest in the Sierra Leone context, but are definitely forest under global definitions (see Figure 2). Figure 8 illustrates the extent of land that the FAO Country Reports will describe as "forest" using the 1975 land cover data. But note that this is still a significant under-estimate of what is recorded in the Country Report as it does not depict the 50% of farm-bush that is included as forest "where it is not dominated by agriculture".

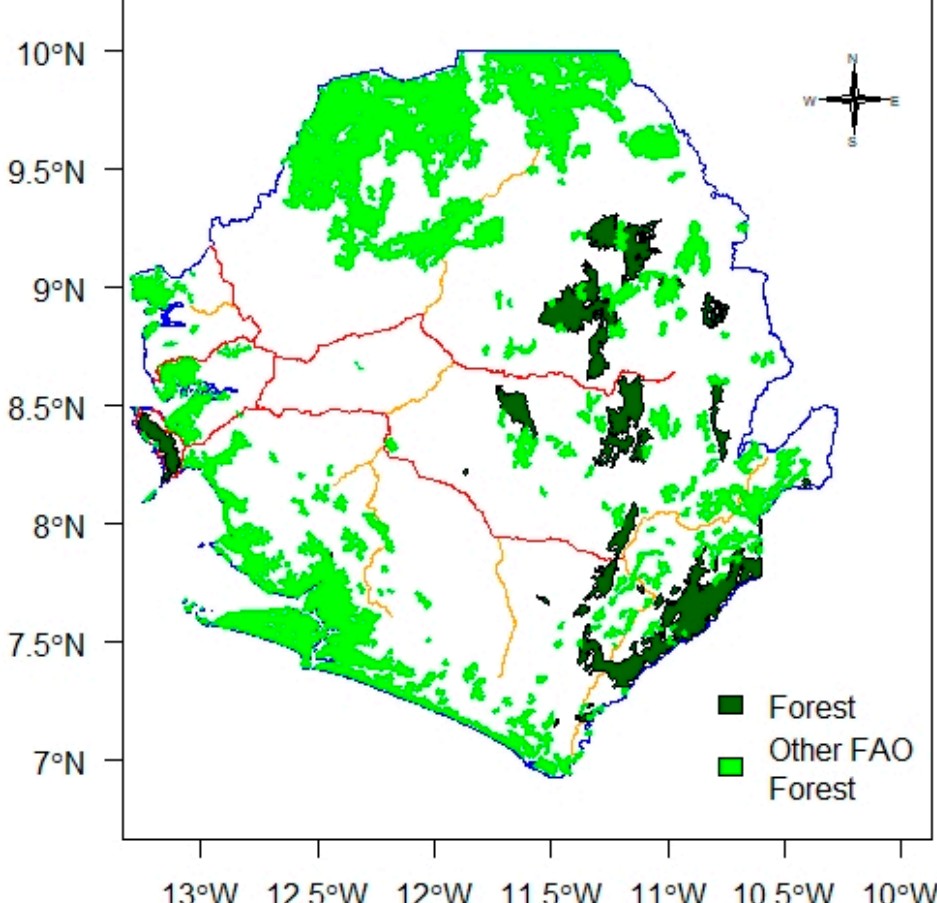

**Figure 8.** FAO Global Forestry Assessment classes applied to the 1975 land cover map. ("Forest" is; primary forest, coastal forest, rubber plantations and mangrove, "Other FAO Forest Classes" is; secondary forest, fringing flooded forest, and densely wooded savanna. Not included on the map is Raphia palm swamp and the 50% of bush-fallow land not considered as under intense agriculture).

While the majority of conservation efforts are concentrated on protecting forests there are various types of woodland (forest under global definitions) that are under threat but neglected. Woody savanna covers large areas of northern Sierra Leone and to a lesser amount along the coast (Figure 9). In the densely wooded savanna (described in the "1975" monograph [29] as "Woodland with fairly closed canopy; trees up to 15 m tall with an undergrowth of tall grasses up to 3 m tall"), the major threat appears to be the unregulated harvesting of *Pterocarpus erinaceus* for export. There is already some tension between arable and livestock farmers in this area and the risk that timber harvesting plus fires set by the partly nomadic cattle herdsmen may change the environment. Areas of *Lophira* savanna that are close to roads (< 2km) are being heavily harvested for charcoal, and in some locations, this seems to have caused an increase in grassland and the invasion by other exotic and overabundant species, such as *Imperata cylindrica* and *Chromolaena odorata* sustained by frequent annual fires, impeding the regeneration of *Lophira lanceolate*. Figure 9 illustrates the extent of "woody" savannas in 1975, but quantitative data are lacking.

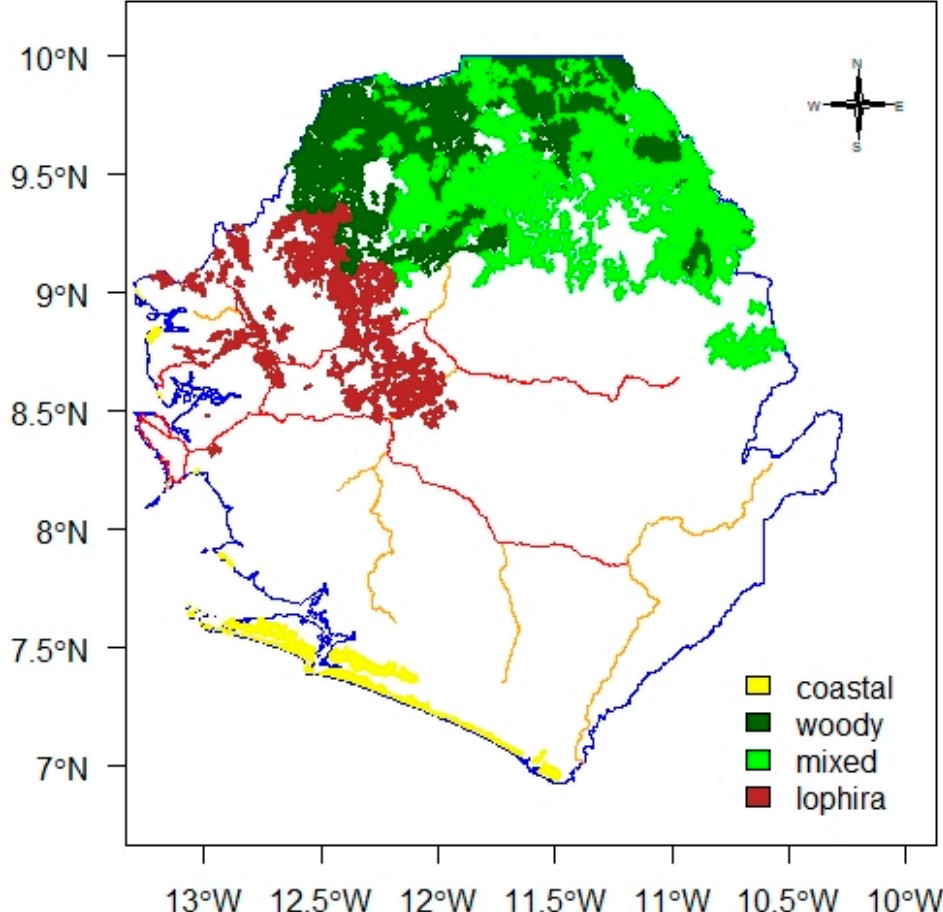

**Figure 9.** The extent of the savanna woodlands in 1975.

## 4. Discussion

Few would argue that tropical forests are not important; they offer a wide range of ecosystem services, including carbon sequestration, moderation of air and water quality, provisioning services and cultural services, as well as acting as biodiversity hotspots. Sierra Leone is one of the countries in the globally important Upper Guinea Forest zone [39] and while Liberia and to a lesser extent Cote d'Ivoire retain significant areas, the trees are under threat from logging and mining and wildlife from the "bush-meat" trade, forests may also be under threat from climate change [40]. While there have been many attempts at valuing ecosystem services, since Costanza's influential paper [41], the first and most critical step is to try and estimate how many fragments are left and how they are threatened.

"Forest" may be considered as land use or a land cover, if it is a land use then it may or may not have trees on it, if it is a land cover then it has trees; the question is how many? Comber et al [11] discuss this semantic confusion as to whether "forest" is a use of a cover in some detail. As part of the discussions around the Kyoto Protocol, Lund [25] identified countries that defined forests in terms of biophysical characteristics (shown here as Figure 1), and it is worth noting that not only is there little consistency but that many countries do not define (legal) forests in this way. Of the freely available global data sets, Global Forest Watch, uses MODIS data with a pixel of 6.25ha and any pixel with more than 30% tree cover is a forest, but, the Global Forest Resource Assessment specifies a much lower area (0.5 ha) and a much lower canopy cover (only 10%). To add to the confusion, the Country Reports for Sierra Leone [1] extrapolate from a survey conducted in 1975 and aggregate habitats not normally considered forest in Sierra Leone. Other global data sets, e.g., Hansen et al [26] avoid defining "forest" and merely report percentage tree cover where trees are anything over five meters (how tree height is derived from spectral data is unclear).

The dominant narrative is that once Sierra Leone was almost completely covered in forest and that the deforestation has been rapid and extreme. The idea of continuous forest cover over most of Sierra Leone is disputed by some, e.g., Fairhead and Leach [5–7] and among those that accept the "primal forest" idea, deforestation could have been substantially completed by the beginning of the 20th Century [15,16] or the forest could still have been largely intact until the 1960s [18]. In an attempt to throw some light on this subject we have analyzed three maps of Sierra Leone covering a 50-year period. In the two later maps [29,30] "forest" is definitely a land-cover, and the implication is that it is in the third ("1950") map. As the civil war displaced perhaps half the population, it can be expected to have had a major impact on land use and land cover, but the effect is mixed; fear of rebels allowing some forests to regenerate, while the rapid doubling or tripling of other communities by IDP (internally displaced person) caused local deforestation. Studies correlating rates of deforestation / reforestation with known conflicts due to the war, such as Reference [4], are hampered by the use of international definitions of forest, such that the paper is actually reporting levels of agricultural activity. Despite the narrative of rapid and extensive forest loss, most of the forests that were identified and mapped in the 1950s are still in existence. While the boundaries of the forest vary over time, much of this variation is probably due to the way that forests were defined and the technology available to map them. The greatest loss of forest is that of the Tonkolili Forest which has almost entirely disappeared. All forests are under threat from mining, with the northern part of Western Area Forest under severe pressure from uncontrolled urban expansion, and the likelihood that once the coastal ring-road is completed, further urban expansions in the south are expected. Gola, like the other forests, has legal protection, but in Gola this is backed up by serious inflow of funds from outside the country and more recently from payments from REDD+ (Reducing Emissions from Deforestation and Forest Degradation). Forests like Loma are to some extent protected by the poor road infrastructure and inhospitable terrain making commercial exploitation uneconomic.

## 5. Conclusions

There appears to be no legal definition of forest in Sierra Leone; neither the Biodiversity Action Plan nor the draft Forest Policy Document defines what they mean by "forest". Statistics reported at five-year intervals by the FAO in the Country Reports for the Global Forest Resource Assessment exercise reflect definitions and aggregations that are not commonly used in Sierra Leone. Without a clear agreement on what a forest is, any statistics on amount or change (whether positive or negative) are impossible to interpret. At the moment, it is possible to defend any figure for the area of "forest" from less than 5% to 75% of the land area. International definitions used in global estimates of forest and tree cover are not particularly useful in the Sierra Leone context as they aggregate too many distinct vegetation forms.

We would argue that as the last national forest inventory was in 1975, we are overdue for another one. A great deal of this could be done with remote sensing, if and only if, "forest" is defined in terms of land cover and not as land use. There are a number of novel sensors, such as Lidar (light detecting and ranging) and SAR (synthetic aperture radar) that offer the promise of detailed land cover and biophysical parameter estimation (such as, vegetation height and above ground biomass) [36–38]. There are, however, several technical issues, such as the "saturation" of estimates of above ground biomass in mature forests. There may also be economic issues in that data that can be obtained free for "research purposes" may be unaffordable when required for a national survey. A new national inventory will first need to determine:

- Who decides what categories and parameters should be mapped (who are the *de-facto* and who are the *de-jure* stakeholders)?
- What decisions are expected to be made using the data (for example, allocation of land for industrial scale plantations, above ground biomass for compliance with REDD+, examining policies to improve resilience to rapid climate change, biodiversity condition assessment of protected areas, etc.)?

- How are classes or categories going to be reliably distinguished (and what is the required accuracy and precision that is needed for the decisions being made with the product)? What is an appropriate MMU (minimum mappable unit) given the decisions that are to be made with the data?

The above decisions need to be documented in the metadata as they have a profound impact on what can and cannot be done sensibly with the data [42,43]. Any results obtained from remote sensing will still need to be carefully checked against what observers see on the ground, but note that what is seen on the ground is not the "truth" in some objective sense, but an alternative interpretation. There are some immediate technical issues defining "forests" (and other land covers) in Sierra Leone:

- The presence of mixed plantations of coffee, cocoa, cola nut, and fruit trees, as these mixtures are spectrally and structurally very similar to natural forests (both consist of mixed species with an uneven canopy). Some (see Reference [7]) consider they are "forest" whilst others [30] are clear they are not. Such mixed plantations may also be adjacent to sacred groves which are close to many small rural settlements and although theses seem to be reduced in extent, they are known to be botanically interesting [44] even if inaccessible to the uninitiated.
- Are mangrove swamps "forests"?
- How should "bush-fallow" be characterized? It covers as much as half the country, but as virtually no land is fallowed for more than 10 years (given the burgeoning population) it seems excessively optimistic to refer to it as "regenerating forest". While the terms shifting cultivation and slash-and-burn are frequently heard, as a land-use it would be more accurate to consider bush-fallow a part of a sustainable agroforestry system (one where the trees and the crops are separated temporally not spatially). Regenerating forest is not part of an agricultural system. In terms of land-cover, bush-fallow is characterized by even stand age and a relatively impoverished suite of species (plants regarded as pioneers and respond well to what is effectively coppicing). After 10 years the structural and species diversity for trees becomes more pronounced, although two species, *Anisophyllea laurina* and *Pentadesma butyracea*, can become increasingly dominant to the exclusion of most species in the northern and southern regions of Sierra Leone. Traditionally, the bush was considered mature enough to farm only when certain species (such as *Musanga cecropioides*) became prominent, but increasingly land is farmed again before this stage is reached.
- How to characterize the different types of savanna: How many are there (four were mapped in 1975), and are they "woodland" or "forest"? As it seems possible that the density of trees is declining due to logging and wildfires, can this change be detected and should it be recorded?

**Author Contributions:** Conceptualization, A.L. and R.W.; Methodology, R.W.; Software, R.W.; Validation, A.L. and R.W.; Formal Analysis, R.W.; Investigation, A.L. and R.W.; Resources, R.W.; Data Curation, R.W.; Writing-Original Draft Preparation, R.W.; Writing-Review and Editing, A.L.; Visualization, R.W.; Supervision, A.L.; Project Administration, A.L. and R.W.; Funding Acquisition, not applicable.

**Funding:** This research received no external funding.

**Acknowledgments:** We would like to thank colleagues for encouraging us to put these ideas on paper.

**Conflicts of Interest:** The authors declare no conflict of interest.

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
