# Peer review of "What Happened to the Forests of Sierra Leone?"

_land, doi:10.3390/land8050080_

Round 1

Reviewer 1 Report

The authors have improved this paper significantly. It will be good to see a paper published about Sierra Leone, from an Sierra Leonean institution.

Author Response

Dear Reviewer,

Thank you for your support. We have reread the paper and made a few minor corrections to the langueage as suggested by you.

Sincerely

Richard Wadsworth and Aiah Lebbie

Reviewer 2 Report

I appreciate the authors’ response to my comments. 

1)      Their response in the comments, rather than the text itself, reacting to my first point presents a good summary of the motivation for the study, how it would challenge the conventional wisdom of deforestation rather than merely provide a new data point for measuring deforestation.  This motivation should be more up-front, in both the abstract and the introduction. 

2)      Thank you for improving the maps.

3)      Thank you for the clarification.

4)      I appreciate the clarification regarding charcoal.  However, you do not, in the conclusion, address the definitional question of how in terms of land cover, bush-fallow and regenerating forest will seem identical, yet in terms of land use, they are quite distinct.  This should still be addressed.

5)      Thank you for removing the footnote.

A couple issues with newly added material:

Figure 7 should have a legend.

Figure 8 has “Forest” and “Other FAO Forest Classes” in the legend, yet the caption lists a series of forest types—primary forest, secondary forest, etc.—without specifying which is in “Forest” and which is in “Other”.  Please clarify this.

Author Response

Dear Reviewer,

Thank you for your comments and help with this paper.

1. abstract and introduction strengthened and expanded by inclusion of some of the ideas that we put in the comments to you, but not in the earlier draft of the paper.

2.,3, 5 agreed

4. more details have been added about the effect of charcoal and the characterisation of bush-fallow. In terms of land cover what is important in these two cases is the species composition and the possible invasion by noxious weeds or the shortening of the fallow to an extent that prevents the characteristic suit of species developing.

Additional maps: Figure 7 now has a legend. The caption for Figure 8 has been revised

Reviewer 3 Report

I already reviewed this paper in the past, reporting a positive feedback for it. I can appreciate that some improvements have been made, so I just recommend a summary copy editing and some follow up on the conclusions and improve the debate on future possible research gaps. 

Author Response

Dear Reviewer,

Thank you for your help with this paper.

We have strengthened the abstract and introduction to provide more of a description of the motivation of the study and also strengthened some of the conclusions and results.

Sincerely

Richard Wadsworth and Aiah Lebbie

Reviewer 4 Report

The paper presents a quite interesting study highlighting the challenges surrounding understanding "forests," and changes to them, in Sierra Leone, and calls for an updated survey. 

I think the paper is in great shape for publication in Land. However, I suggest that the manuscript requires a careful proofread for grammar and punctuation. For example, consider the following selected errors from page 2: 

line 54: "thesis" is singular--should be "theses" if plural

line 68: use a comma before coordinating conjunctions when separating two independent clauses

line 69: "it's" is a contraction for "it is." In this case, should be "its," which is the possessive form of "it"

line 75: commas and periods go inside the quotation marks. 

Such grammatical and punctuation errors occur throughout the document, and ought to be addressed before publication.             

Author Response

Dear Reviewer,

Thank you for your help with this paper.

We have gone through attempting to catch all the grammatical inconsistencies and errors.

Sincerely

Richard Wadsworth and Aiah Lebbie

This manuscript is a resubmission of an earlier submission. The following is a list of the peer review reports and author responses from that submission.

Round 1

Reviewer 1 Report

GENERAL COMMENTS

One sees too few scientific papers in the environmental sector from countries like Sierra Leone with its small cadre of scientists, grave public sector issues and post-Civil War and Ebola recovery efforts. What the authors produce is the basis of a potentially important paper that Land would be well advised to publish. 

However, these quite a few improvements are required. Some of my general comments overlap with detailed comments below; they are that:

(1) The contextual introduction is generally good but it could be expanded in terms of the early material (no mention of coastal trading posts and the SL railway) or the differences between Guinea savanna and closed forest, (2) the affects of the Civil War (and any Ebola-related affects), and (3) nothing on the underfunded government departments in the natural resources area. 

(2) The data sources and menthol used need some more details

(3) The results section is a series of maps and accompanying descriptions. However, the maps are difficult to read in some cases and key information is not included (suggestions given in the detailed comments). Some maps need better descriptions of what can be seen (most readers of the paper will not know where places are for instance) and for some figures as discussion of what has caused changes is given. In this section the authors should stick to reasonably detailed descriptions of the maps/figures.

(4) The discussion is inadequate as it is almost entirely a plea for a new Forest Inventory. I believe that is the conclusion of this research and should be moved a revised Conclusions section or a separate section before the conclusion. That leads a gap where (i) you need to develop a discussion section in which the actual processes on the ground and the technical concerns that combine to produce each map need to be discussed, and (ii) gather the evidence that leads to you plea for a new inventory.

I fully encourage the authors to work on these revisions. This paper should be published but not in its current form.

DETAILED COMMENTS

Lines 62-6.I find the part of this sentence that refers to the 1920s,40s and 60s confusing. Suggest a rewrite of the sentence to make clear if the 'dominant narrative' was being used to describe deforestation in these three decades. I am also having some difficulty in thinking ability whether the references were being applied to the whole of the protectorate. If my memory serves me correctly even the early 1900s reference like Lane-Poole really only argued that deforestation was proceeding at a fast rate in some areas (along the railway for example). Little mention was made of the woodlands of the Guinea savanna in the north and north-west.

Line 105 While I have no issue with Fig 2, its description as a crude thematic needs to be changed.

Lines 113-115. Some more details of the methodology is required.

Lines 117-8 Surveys and Lands Department. Maybe some context as to what the Surveys and Land Department was and what it covered (general cartographic survey? agriculture? forestry?) and, in particular, what was the 1950s Soil Conservation Team (upper or lower case, unsure)

Line 119. This should be referenced in the bibliography. But I know hard copy documentation in SL was problematic well before the Civil War. I assume it was a map in a report, if the so the report should be referenced. The point about referencing these documents applied to the materials used in source #2. I do know that report exists. 

Lines 141-144. This paragraph leads me to conclude the authors of this paper attempted this GE classification, but readers need to be told exactly that if it is true. Another point(s) that I'd like to see made clear would be if the forest areas from GE imagery were used at all? And if they were/were no maybe be a bot more explicit about the types of 'errors' you (if it was the authors) encountered.

FIg 3. I think that map should be larger in the final published paper as it is too small to see the detailed changes - which are mainly around the edges - of these areas of forest. I am not sure the cross hatching symbols are the best way of visualising these data, but for now I cannot think of a better way. I also think some 'geographical' information would be useful, say the major towns/cities, and of course scale, lat long info and a north arrow. A better title might be something like 'Major forested areas in Sierra Leone, 1950, 1975 and 2002'

Figs 4, 5 and 6. I make the same points about the need to include 'geographical' information on this map, and note Tall Trees in the legends should be 2002 not 2000. In addition I would like to see some consistency for titles for these three maps (see comment on Fig 3 for guidance. In Fig 6 I do not recognise the term 'Midlands' as a generally accepted term for this part of the country, in fact the middle of SL in only the left-hand margin of this figure; perhaps a more recognisable term should be used. The ellipse is a poor way to illustrate the loss of forest in those particular blocks, perhaps names would be better or even an inset map that blows then up? In addition in these three figures (4-6) a smart idea would be to label the forest blocks in them so readers could refer back to Figure 3. My comments about geographical information apply to Figs 7, 8 and 9 (and the titles needs some attention).

Lines 175-6. This phrasing of this sentence could be improved if rewritten.

Lines 175-183. You mention that the Kambui area tells a different story (to the Gola), but you do not tell the reader what the story of the Gola is. This makes the comparison you refer too impossible of the reader to make. So, if there are two contrasting stories for these tow forests, they m=need to be detailed in this paragraph.

Lines 186-8. The narrative of Fig 6 in these three lines tells us little and needs expanding.

Lines 192-5. I do not think that paragraph (or Fig 7) are in the best place in this manuscript. Please consider the point you are making, expand the paragraph and re-locate. 

Line 214. Gola North is an undefined area (see Fig 3)

Line 216, you write 'estimates are quite close'. So far we have seen maps of forest areas at different times (some of these are not easy to read), but isn't one of the points about GIS analysis that you can generate numerical estimates of areas. It occurs to me at this point that nowhere have you produced tables of areas for the three years (or for the Hansen/tile comparisons in this paragraph. Tables of numerical area data for each forest area would surely be useful in the results section.

Line 227. The paragraph starting in this line is a big switch of focus, especially as none of these savanna woodland areas are on Fig 3. You are conflating forest and savanna woodland in the readers mind. Some clearer guidance is needed at the start of this paragraph. Then, before you really even describe the general situation for these woodlands, you focus in on one species and talk about some local geography (and locations) which are not indicated on any map (I have no idea where these places are), indicate a data lacunae, suggest a way the data could be gathered, and talk about a lorry count without giving any details. Thats all a bot confusing and unsatisfactory, especially as most of what you write does not relate to Fig 9.

Line 243. Can you quantify thus spatial autocorrelation?

Line 245. We now find Tonkolili and Nimi have been systematically cleared, (i) should have mentioned that in Results section, and (ii) discussion this systemtic clearance in Discussion (who?, when? how? what do you mean by 'systematic').

There is a very big content and conceptual jump from paragraph 1 in the Discussion to paragraph 2. And in fact the remains of the Discussion section is not really a discussion but a justified plea for a new Forest Inventory. Shouldn't that really be a conclusion?

There are a few issues with the bibliography, please check carefully,

Reviewer 2 Report

This manuscript presents a summary of forest changes in Sierra Leone since 1950 using three cartographic data sources, concluding with the need to conduct a new forest inventory.  As such, it comes across more as a commentary or position paper instead of a standard research article.  This is the rationale for the low ratings in the multiple choice section.  There is no clear research question beyond a description of how forest cover has changed.  Because of this, while I have no doubt that those in Sierra Leone need and can use a new forest inventory, this limits the significance of the content and interest to the readers largely, albeit not exclusively, to those with interests specific to Sierra Leone.

Should the manuscript get published, the maps would benefit from some improvements and additional details. In many cases, there are landmarks referenced in the text which the authors seem to assume are locations everyone would know, such as Makeni and the Loma Mountains for Figure 3.  By and large, if a location is important enough to have in the text, it is also important enough to label on a map.  Another issue is that Figures 4-6 seem to be zoomed in on screen captures for previous maps; the spacing between lines in the hachure shading increased, implying it is a zoomed in portion of the previous map, not a new map with the same symbology and, presumably, more detail.  

This is a small detail in the methodology; you write on line 210 that you generate an “average value” for each pixel’s neighborhood.  Is this the average of the initial estimated tree cover, or the average of the fuzzy logic probability? 

Furthermore, some of the recommendations and discussion items are presented without much evidence from the results.  You assert in lines 236-238 that deforestation is occurring because of charcoal production, with a shift to permanent grassland, especially near roads.  However, this is presented without roads at all, or any indication that the cause of deforestation was studied, instead of recording deforestation.  Also, you distinguish between “bush-fallow” and “regenerating forest” within your work, yet the two are spectrally similar.  While this distinction may be useful from the perspective of land use, it may nonetheless be similar from a land cover perspective.

Lastly, it’s a small thing, but there’s no reason for footnote 1. 

Reviewer 3 Report

The paper entitled “What happened to the forest of Sierra Leone” presents an assessment of mayor land use changes using three maps (1950, 1975 and 2000) to describe the forest loss and ask the urgency of a new forest inventory.

Even if the issue is of great interest and debate on this field is flourishing I found this article a bit odd for this journal. Despite the style, there are substantial flaws in the methodology. It seems that authors didn’t get in touch with the basic assumption of land use change analysis that is the comparability between land cover maps. Therefore, the estimation of changes has to be done with cross tabulation or differential analysis ones that land use maps are comparable in different time thresholds. This paper mixes considerations using ancillary data from bibliography and maps (which are clearly uncomparable).

Please see these references

Benini, L., Bandini, V., Marazza, D. and Contin, A. (2010) ‘Assessment of land use changes through an indicator-based approach: A case study from the Lamone river basin in Northern Italy’, Ecological Indicators, 10(1), pp. 4–14. doi: 10.1016/j.ecolind.2009.03.016.

Verburg, P.H., Schot, P.P., Dijst, M.J. and Veldkamp, A. (2004), “Land use change modelling: current practice and research priorities”, GeoJournal, Vol. 61 No. 4, pp. 309-324.

Even the part of the manuscript dedicated at the definition of forest sounds a bit strange since rather than the “definition” it is important to know which is the “detection factor” that constitute the base to classify a forest. But nowadays many land use dataset are provided and I don’t think this issue represent a real obstacle in for land use classification.

Therefore I found the methodology too general, whithout a proper introduction of the land use change assessment methodology (how did you compared the maps? Which method of analysis, which software? How did you quantify the cover?). The discussion mixes general statement from a documental analysis and statements from the comparison of forest maps. Images are of too low resolution to make a proper evaluation. Quotations in the texts are wrong by a stylistic point of view (see line 165: you cannot write a phrase like this “[34] also provides… if you want to refer to a specific author you have to write “Peterson in his work of 1995, founded that… [34].”

If you don’t have enough data to integrate your analysis (or you cannot apply a proper land use change analysis) I encourage you to resubmit to another journal.

Reviewer 4 Report

I would not use a question as the title of a paper. Nonetheless, the study looks interesting and challenging, trying to lift up a debate on a clear definition of forest.

As a recommendation to be given, I would suggest the authors to focus more on possible passages, especially in the discussion or in the conclusion, which could contest the weakest points of the international definition, and maybe to propose some new keys of interpretation, particularly useful for the case of Sierra Leone and derived directly from the results of their study.

In this sense a brief summary of results with the correspondent resulting inadequacies of the International definition of forests could be provided in the discussion section.

It would strenghten the external validity of this study.

Moreover, to give a more economic perspective, I would suggest to stress on the important role of forests as a crucial carbon sink, but also as a “tool” of natural insurance to prevent ecological disasters. For instance, you may check for example:

Valente, D., Miglietta, P. P., Porrini, D., Pasimeni, M. R., Zurlini, G., & Petrosillo, I. (2019). A first analysis on the need to integrate ecological aspects into financial insurance. Ecological Modelling392, 117-127.